# Apolipoprotein E regulates the maturation of injury-induced adult-born hippocampal neurons following traumatic brain injury

Yacine Tensaouti[1], Tzong-Shiue Yu[1], Steven G. Kernie[1,2]*

1 Department of Pediatrics, Columbia University Vagelos College of Physicians and Surgeons, New York, New York, United States of America, 2 Department of Neurology, Columbia University Vagelos College of Physicians and Surgeons, New York, New York, United States of America

* sk3516@cumc.columbia.edu

**Data Availability Statement:** All relevant data are within the paper and its Supporting Information files.

**Funding:** This work was supported by the National Institutes of Health grants R01 NS095803 (sgk)

## Abstract

Various brain injuries lead to the activation of adult neural stem/progenitor cells in the mammalian hippocampus. Subsequent injury-induced neurogenesis appears to be essential for at least some aspects of the innate recovery in cognitive function observed following traumatic brain injury (TBI). It has previously been established that Apolipoprotein E (ApoE) plays a regulatory role in adult hippocampal neurogenesis, which is of particular interest as the presence of the human ApoE isoform ApoE4 leads to significant risk for the development of late-onset Alzheimer's disease, where impaired neurogenesis has been linked with disease progression. Moreover, genetically modified mice lacking ApoE or expressing the ApoE4 human isoform have been shown to impair adult hippocampal neurogenesis under normal conditions. Here, we investigate how controlled cortical impact (CCI) injury affects dentate gyrus development using hippocampal stereotactic injections of GFP-expressing retroviruses in wild-type (WT), ApoE-deficient and humanized (ApoE3 and ApoE4) mice. Infected adult-born hippocampal neurons were morphologically analyzed once fully mature, revealing significant attenuation of dendritic complexity and spine density in mice lacking ApoE or expressing the human ApoE4 allele, which may help inform how ApoE influences neurological diseases where neurogenesis is defective.

## Introduction

Among all trauma-related insults, traumatic brain injury (TBI) represents the highest contributor to death and disability globally, remaining a major public health concern worldwide [1, 2]. Individuals suffering from traumatic brain injuries can spontaneously recover to some degree, suggesting the existence of innate repair mechanisms. One such potential mechanism is hippocampal injury-induced neurogenesis [3]. Mouse modeling of TBI leads to neuronal cell death in the dentate gyrus of the hippocampus with loss of both mature and immature doublecortin-expressing neurons [4, 5]. It has been shown that injury-induced activation of resident neural stem/progenitor cells (NSPCs) from the subgranular zone of the dentate leads

and R56NS089523 (sgk) who played no role in the study design, data collection and analysis, decision to publish, or preparation of the manuscript.

**Competing interests:** The authors have declared that no competing interests exist.

**Abbreviations:** ApoE, Apolipoprotein E; CCI, Controlled Cortical Impact; iGCL, Inner Granule Cell Layer; KO, Knock-Out; NSPCs, Neural stem/progenitor cells; oGCL, Outer Granule Cell Layer; RT, Room temperature; TBI, Traumatic Brain Injury; WT, Wild Type.

to neurogenesis that compensates for the loss of these vulnerable cell populations [5, 6]. These injury-induced neurons functionally integrate into the pre-existing circuitry [7] and appear to be necessary for at least some aspects cognitive recovery in mice as ablation of injury-induced hippocampal adult-born neurons impairs cognitive recovery after TBI in mice [3]. It has also been reported that these new neurons have morphologic aberrancies that may increase seizure susceptibility and contribute to other aspects of brain injury pathophysiology [8]. Thus, it remains unclear what the overall effect of injury-induced neurogenesis following TBI might be, though its relevance to the overall adaptation for the hippocampus to traumatic brain injury is clear [7, 9–13]

Alterations in adult hippocampal neurogenesis is a hallmark of hippocampus-associated neurological and neurodegenerative diseases, inducing cognitive deficits (e.g. difficulty in learning new information, memory loss) as observed in major depressive disorder, schizophrenia, Alzheimer's disease, epilepsy, and TBI [14–19]. Understanding how adult neurogenesis is regulated is therefore essential before manipulation of dentate gyrus progenitor cells can be a viable therapeutic strategy in promoting neuronal regeneration. However, despite ever-expanding interest around the incorporation of adult-born granule cells to the hippocampal circuitry, the mechanisms directing this dentate gyrus neurogenic response remain poorly understood. One key regulatory gene for adult neurogenesis is Apolipoprotein E (ApoE), which has been shown to negatively regulate postnatal proliferation of dentate gyrus NSPCs [20, 21], while ablation of ApoE expression shifts NSPC differentiation towards astrogenesis instead of neurogenesis [22].

ApoE is primarily of astrocytic origin though it is also expressed by Type I neural stem cells before they develop into more mature neurons [17, 21]. ApoE is the main lipid carrier in the brain and regulates its transport and homeostasis, which in turn is critical for supporting neuronal development, beta-amyloid metabolism, and blood-brain barrier integrity maintenance [23–26]. In humans, ApoE is found in three commonly occurring protein alleles with ApoE3 considered the "wildtype" form as it is most predominant in humans and is Alzheimer's disease risk-neutral [27]. ApoE2 is the rarest allele in the population and is thought to have neuroprotective effects [28], while ApoE4 occurs in approximately twenty percent of the population and is the greatest known genetic risk factor associated with the development of late-onset Alzheimer's disease [29]. Moreover, ApoE4 has also been associated with poorer outcomes after TBI, including emergence of dementia and cognitive decline [30–33], and is also associated with small vessel disease and cognitive impairment on a vascular basis i.e. hypoxic-ischemic white matter damage and resulting dementia [34]. ApoE4 is also linked to a greater incidence of moderate or severe contusions [35] as well as concussions [36]. ApoE4 genotype combined with TBI is thought to increase the risk of developing Tauopathy and Alzheimer's Disease [37] as well as post-traumatic epilepsy [38] and impaired spontaneous blood brain barrier repair [24]. Neurogenesis is affected in an ApoE isoform-dependent manner after both ischemic stroke [39] and controlled cortical impact [17] in mouse models but not after concussion [40]. Recently a large meta-analysis in humans concluded that outcome following TBI occurs in an ApoE isoform-dependent manner thus validating its importance in overall TBI, but the underlying mechanism remains entirely unknown [41].

ApoE regulates proliferation of NSPCs in both the healthy and injured mouse brain, and is influenced by human ApoE isoforms [17]. Moreover, ApoE deficiency and ApoE4 are both detrimental to dendritogenesis of adult-born granule neurons in the intact mouse brain [42], leading to significantly decreased dendritic complexity and spine density. In this study, we explored the effects of ApoE-deficiency and ApoE4 on the neuronal development of injury-induced adult-born cells.

## Material and methods

### Animals

All experimental procedures were in accordance with the Guide for the Care and Use of Laboratory Animals of the National Institutes of Health and approved by the Institutional Animal Care and Use Committee at Columbia University (Protocol Number: AC-AAAT5462). Experimental animals were humanely housed and cared for under the supervision of the Institute of Comparative Medicine at Columbia University. All survival and non-survival surgeries were carried out under general isoflurane anesthesia, and animal suffering was minimized as further described. C57BL/6J (RRID:IMSR_JAX:000664; WT) and ApoE Knock-Out mice (B6.129P2-Apoetm1Unc/J; RRID:IMSR_JAX:002052) were both purchased from The Jackson Laboratory. ApoE3 (B6.129P2-Apoetm2(APOE*3)Mae N8; RRID:IMSR_TAC:1548) and ApoE4 (B6.129P2-Apoetm3(APOE*4)Mae N8; RRID:IMSR_TAC:1549) humanized mice were purchased from Taconic.

### Controlled cortical impact injury and retroviral injections

Sixteen mixed-sex WT, ApoE-deficient, ApoE3, and ApoE4 mice (2 males and 2 females in each condition) underwent CCI at six-weeks of age as previously described [3]. Twenty minutes following intraperitoneal injection of analgesics (5 mg/kg Carprofen–Rimadyl), mice were placed in an induction chamber and subjected to general anesthesia using 4% isoflurane supplemented with 1L flow of oxygen. Head fur was clipped and animal heads were fixed on a stereotactic frame where anesthesia was maintained throughout surgery using a nose cone linked to the Isoflurane Evaporator (Summit Anesthesia Solutions). The isoflurane dose was progressively reduced from 4% to 2%, and the level of anesthesia was assessed using toe pinch while internal temperature was kept at 37˚C via a heating pad (Adroit, Loudon, TN). After the scalp was disinfected using three separate antiseptic swabs (B10800 –Prevantics), local analgesics (2mg/kg Bupivicaine–Hospira) were subcutaneously injected and ophthalmic ointment was applied (one drop/eye). Following a scalp midline incision, the soft tissues were reflected and a 5 mm by 5 mm craniotomy was performed between bregma and lambda (to the left of the sagittal suture) to expose the underlying dura. Single moderate cortical contusion injuries (3 mm stainless steel impounder tip; speed: 4.4 m/sec; deformation: 0.7 mm; dwell time: 0.3 sec) were delivered with the Leica One Stereotaxic Impactor device (Leica, Houston, TX). Immediately after injury, a retrovirus carrying enhanced green fluorescent protein (eGFP) was stereotactically infused into the mouse dentate gyrus (Moloney Murine Leukemia Viral vectors were generated by the GT3 Core Facility of the Salk Institute). Retroviruses only infect mitotic (actively dividing) cells in a stochastic manner. As a result, only neural stem and progenitor cells from the neuronal lineage express GFP in the infected dentate gyrus. Other potentially dividing cells such as reactive astrocytes and microglia can morphologically be differentiated without ambiguity. The constitutive expression of eGFP in infected NSPCs allowed for permanent labelling and further dendritic arborization tracing. As previously described [42], to capture NSPCs from both the dorsal and ventral parts of the dentate gyrus and from both hemispheres, one microliter of packaged retrovirus [$1^*10^9$ transducing units (TU)/mL)] was injected at the rate of 0.1µl per minute as: Antero/Posterior = -2.0mm & -2.5mm, Medio/Lateral = ±1.55mm & ±2mm, Dorso/Ventral = -2.0mm & -2.25mm all coordinates from bregma, using a micro infusion pump (KD scientific) linked to a 10µl Hamilton syringe (Model #801), for a total of 4 injected sites per animal. After surgery, the scalp was closed with sutures, topical antibiotic ointment was applied (one drop of Neosporin) and animals were placed in their cages and allowed to recover from anesthesia.

## Tissue processing and immunohistochemistry

Since it takes approximately 4 weeks for adult born granule cells to mature and integrate into the trisynaptic circuitry [43], we chose this post-surgery time point to perform our histological analysis. Transcardiac perfusion was performed with 150 mL of 4% paraformaldehyde (PFA) in 1x PBS delivered via a peristaltic pump at a rate of 10ml/min (MasterFlex L/S, Cole Parmer); animals were subjected to general anesthesia during perfusions (Isothesia Isoflurane, Henry Schein Animal Health). Post-fixation was achieved by incubating whole brains overnight in in 4% PFA/1x PBS before sectioning of serial 50μm coronal brain slices (vibratome VT1000S, Leica). Samples that encompassed the hippocampus were all sequentially collected in six Eppendorf tubes with either 1x PBS or antifreeze solution (30% Glycerol/30% ethylene glycol in 1x PBS) and free-floating sections were used for further immunohistochemistry while the remaining samples were preserved in the dark at -20˚C. Samples were first rinsed with 1x PBS to eliminate antifreeze solution when applicable (3x5min). Then a standard immunostaining procedure was performed at room temperature (RT): permeabilization step [0.3% Triton X-100 in 1x PBS (PBST; 3x10mn)] followed by blocking step [PBST with 5% Normal Donkey Serum (NDS, Jackson ImmunoResearch Labs, Cat# 017-000-001; 1 hour)]. Brain sections were then incubated overnight at 4˚C with primary antibodies (1:5000 Goat polyclonal α-ApoE, EMD Millipore, AB947; 1:1000 Rabbit polyclonal α-GFP, Invitrogen, Cat# A-11122; 1:1000 Polyclonal Guinea Pig α-GFAP, Mybiosource, MBS834682) in PBST/5% NDS. The next day, tissues were washed with PBST (3x5mn) and incubated with biotinylated or alexa-conjugated secondary antibodies (1:200 Biotin Donkey α-Goat, Jackson ImmunoResearch Labs, Cat# 705-006-147; 1:200 Biotin Donkey α-Rabbit, Jackson ImmunoResearch Labs, Cat# 711-545-152; 1:200 Biotin Donkey α-Guinea Pig, Jackson ImmunoResearch Labs, Cat# 706-605-148) for 3 hours at RT. Sections were then washed with 1x PBS (3x5mn) and incubated with alexa594--conjugated Streptavidin antibodies (1:200, Jackson ImmunoResearch Labs, Cat# 016-580-084) for 2 hours at RT. Finally, tissue samples were rinsed with 1x PBS (3x5min) before they were slide-mounted and coverslip-sealed (H-1500, Vector Laboratories).

## Neuronal morphological analysis

As previously described [42], upon completion of immunostaining, slides were visualized and imaged with a Zeiss microscope (Axio Imager M2, Zeiss) equipped with a Hamamatsu camera (Orca-R2, Hamamatsu). Interval z-stack images at 1μm were obtained under a 20x objective using an optical fractionator (Zeiss Apotome.2) and stereological image analysis software (MBF Bioscience, RRID:SCR_002526). Acquired z-stacks were then opened with the Neurolucida360 software (RRID:SCR_001775) where 3D neural tracing was performed by a blinded experimenter. Neuronal traces were analyzed using the same software with several morphological parameters of the dendritic arborization chosen for further comparisons (proximal dendritic length before the first division, total cumulated dendritic length, dendritic complexity, dendritic spawn and Sholl analysis) while axons were not studied.

For each animal, a set of brain samples enclosing every sixth section was mounted on slides and endogenous eGFP-expressing dendritic spines were visualized using a Laser Scan confocal microscope (TCS SP8, Leica). Z-stack images at 0.1μm intervals were acquired in the molecular layer of the dentate gyrus under a 63x oil objective with a five-time digital zoom (pixel size = 57.21nm, NA = 1.44, resolution = 512x512, frame average = 4) and were then deconvolved using Autoquant software (RRID:SCR_002465). Z-stacks of dendritic fragments were then visualized in three dimensions using Neurolucida 360 (RRID:SCR_001775) [44], 10μm dendritic fragments were randomly selected for spine counting (no more than one fragment per dendrite) and analyzed by a blinded experimenter. To avoid underestimation of the spine

density that could be due to spherical aberrations over the Z axis, spine counts were expressed relative to fragment lengths [45].

## Statistical analysis

Graphpad Prism (RRID:SCR_015807) was used to perform all statistical analyses. The Shapiro-Wilk was used to assess the normality of data. All results are shown as the mean ± Standard Error of the Mean (SEM) and statistical details are presented in Table 1. Because CCI induces outward migration of adult-born granule neurons [7], we further divided reconstructed neurons from the ipsilateral side into the inner granule cell layer (iGCL, soma located in the inner one third of the granule cell layer) and the outer granule cell layer (oGCL, soma located in the outer two thirds of the granule cell layer), however, this analysis was performed on the ipsilateral side only because outward migration was rarely observed on the contralateral side. Our group previously published the dendritic morphology of adult-born neurons in the uninjured dentate gyrus of WT and various ApoE conditions [42]. We compared these to the contralateral neurons in injured animals analyzed in the present manuscript and found no significant differences. Thus, neurons residing in the contralateral side of injured brains were used as controls in the current study. A Kruskal-Wallis test was used to study non-parametric distributions followed by Dunn's *post hoc* tests and one-way ANOVA followed by Tukey's honestly significant difference (HSD) *post hoc* test was used for parametric distributions. Two-way ANOVA was used to study the effect of different conditions on the Sholl Analysis followed by an uncorrected least significant difference (LSD) Fisher's *post hoc* tests where the Type I error increases with the number of multiple comparisons [46]. Statistical analyses for each individual experiment are summarized in Table 1.

## Results

### GFP-expressing retroviruses track adult-born neuronal dendrite development which are found in close proximity to ApoE-expressing astrocytes

During normal adult hippocampal neurogenesis, NSPCs migrate short distances from the subgranular zone to the inner one-third of the granule cell layer [7, 10, 47]. Typically, adult-generated dentate gyrus neurons migrate to the inner one-third of the granule cell layer (iGCL) and project their dendrites toward the GCL and the Molecular layer, with the first dendritic division occurring around 35μm from the cell body [42]. However, this migratory pattern is not observed after experimental brain injury as CCI causes aberrant migration of newborn immature neurons in the hippocampus with proliferation peaking in the first week and eventually leading to around 50% misplacement of mature injury-induced adult-born granule neurons, many of which become permanently incorporated into the existing dentate gyrus [7, 10, 11]. Because the functional implications of this aberrancy remain unclear, we separately analyzed neurons born in the inner (iGCL) and outer GCL (oGCL) in the ipsilateral injury side, whereas on the contralateral side, we only very rarely observed misplaced adult-born neurons.

Mixed genotype and sex mice underwent moderate CCI at 6 weeks of age immediately followed by stereotactic brain injections of a GFP-expressing retrovirus. Mice were perfused 4 weeks later and brains were removed to assess both quality of the injury and retroviral labelling of mature granule cells before further histological analysis was performed (Fig 1A). Next, sections were stained for ApoE and the astrocyte-specific marker GFAP to determine its expression in relationship to GFP-expressing dendrites where astrocytic expression of ApoE is seen in close proximity with GFP-expressing mature granule cell dendrites by (Fig 1B–1F). We next

**Table 1. Summary of statistics.**

| Table Analyzed | Condition | Statistical Test | p value | DF | R² | Statistic |
|---|---|---|---|---|---|---|
| 1st Branch | WT | Kruskal-Wallis | <0.0001 | | | H (3) = 18.44 |
| | WT: iGCL vs oGCL | Dunn's test | <0.0001 | | | |
| | WT: iGCL vs Contralateral | Dunn's test | 0.5977 | | | |
| | WT: oGCL vs Contralateral | Dunn's test | 0.0073 | | | |
| | WT vs ApoE KO | Kruskal-Wallis | <0.0001 | | | H (6) = 43.62 |
| | WT vs ApoE KO: iGCL | Dunn's test | 0.5543 | | | |
| | WT vs ApoE KO: oGCL | Dunn's test | 0.9172 | | | |
| | WT vs ApoE KO: Contralateral | Dunn's test | 0.2871 | | | |
| | ApoE3 vs ApoE4 | Kruskal-Wallis | <0.0001 | | | H (6) = 46.92 |
| | ApoE3 vs ApoE4: iGCL | Dunn's test | 0.0061 | | | |
| | ApoE3 vs ApoE4: oGCL | Dunn's test | 0.2892 | | | |
| | ApoE3 vs ApoE4: Contralateral | Dunn's test | 0.0227 | | | |
| Nodes | WT | Kruskal-Wallis | 0.5029 | | | H (3) = 1.375 |
| | WT: iGCL vs oGCL | Dunn's test | 0.8527 | | | |
| | WT: iGCL vs Contralateral | Dunn's test | >0.9999 | | | |
| | WT: oGCL vs Contralateral | Dunn's test | 0.9441 | | | |
| | WT vs ApoE KO | Kruskal-Wallis | <0.0001 | | | H (6) = 32.02 |
| | WT vs ApoE KO: iGCL | Dunn's test | 0.0003 | | | |
| | WT vs ApoE KO: oGCL | Dunn's test | 0.006 | | | |
| | WT vs ApoE KO: Contralateral | Dunn's test | 0.0062 | | | |
| | ApoE3 vs ApoE4 | Kruskal-Wallis | <0.0001 | | | H (6) = 46.08 |
| | ApoE3 vs ApoE4: iGCL | Dunn's test | 0.0001 | | | |
| | ApoE3 vs ApoE4: oGCL | Dunn's test | 0.0001 | | | |
| | ApoE3 vs ApoE4: Contralateral | Dunn's test | 0.0003 | | | |
| Total Length | WT | Kruskal-Wallis | 0.1369 | | | H (3) = 0.1369 |
| | WT: iGCL vs oGCL | Dunn's test | 0.187 | | | |
| | WT: iGCL vs Contralateral | Dunn's test | >0.9999 | | | |
| | WT: oGCL vs Contralateral | Dunn's test | 0.2969 | | | |
| | WT vs ApoE KO | Kruskal-Wallis | <0.0001 | | | H (6) = 40.64 |
| | WT vs ApoE KO: iGCL | Dunn's test | 0.0398 | | | |
| | WT vs ApoE KO: oGCL | Dunn's test | 0.002 | | | |
| | WT vs ApoE KO: Contralateral | Dunn's test | 0.0008 | | | |
| | ApoE3 vs ApoE4 | Kruskal-Wallis | <0.0001 | 5 | 0.0862 | F (5, 474) = 8.945 |
| | ApoE3 vs ApoE4: iGCL | Tukey's HSD | <0.0001 | | | |
| | ApoE3 vs ApoE4: oGCL | Tukey's HSD | <0.0001 | | | |
| | ApoE3 vs ApoE4: Contralateral | Tukey's HSD | 0.4031 | | | |
| Angle | WT | Kruskal-Wallis | 0.0157 | | | H (3) = 0.0157 |
| | WT: iGCL vs oGCL | Dunn's test | 0.0134 | | | |
| | WT: iGCL vs Contralateral | Dunn's test | 0.2533 | | | |
| | WT: oGCL vs Contralateral | Dunn's test | 0.6427 | | | |
| | WT vs ApoE KO | Kruskal-Wallis | 0.0005 | | | H (6) = 22.25 |
| | WT vs ApoE KO: iGCL | Dunn's test | 0.2197 | | | |
| | WT vs ApoE KO: oGCL | Dunn's test | 0.0927 | | | |
| | WT vs ApoE KO: Contralateral | Dunn's test | 0.0154 | | | |
| | ApoE3 vs ApoE4 | Kruskal-Wallis | <0.0001 | | | H (6) = 48.8 |
| | ApoE3 vs ApoE4: iGCL | Dunn's test | 0.0006 | | | |
| | ApoE3 vs ApoE4: oGCL | Dunn's test | 0.0907 | | | |
| | ApoE3 vs ApoE4: Contralateral | Dunn's test | 0.0004 | | | |
| Spine density | WT vs ApoE KO | One-way ANOVA | <0.0001 | 3 | 0.4861 | F (3, 258) = 81.33 |
| | Ipsilateral: WT vs ApoE KO | Tukey's HSD | <0.0001 | | | |
| | Contralateral: WT vs ApoE KO | Tukey's HSD | <0.0001 | | | |
| | WT Ipsilateral vs Contralateral | Tukey's HSD | 0.5789 | | | |
| | ApoE KO Ipsilateral vs Contralateral | Tukey's HSD | 0.6437 | | | |
| | ApoE3 vs ApoE4 | Kruskal-Wallis | <0.0001 | | | H (4) = 196.4 |
| | Ipsilateral: ApoE3 vs ApoE4 | Dunn's test | <0.0001 | | | |
| | Contralateral: ApoE3 vs ApoE4 | Dunn's test | <0.0001 | | | |
| | ApoE3 Ipsilateral vs Contralateral | Dunn's test | >0.9999 | | | |
| | ApoE4 Ipsilateral vs Contralateral | Dunn's test | >0.9999 | | | |

*(Continued)*

**Table 1.** (Continued)

| Table Analyzed | Condition | Statistical Test | p value | DF | R² | Statistic |
|---|---|---|---|---|---|---|
| Sholl Analysis | WT: iGCL vs oGCL vs Contralateral | Two-way ANOVA | | | | |
| | | Interaction | <0.0001 | 56 | | F (56, 4727) = 2.651 |
| | | Row Factor | <0.0001 | 28 | | F (28, 4727) = 220.5 |
| | | Column Factor | <0.0001 | 2 | | F (2, 4727) = 12.95 |
| | WT: iGCL vs Contralateral | Fisher's LSD | p < 0.05, 0.05, 0.01 for each 10µm increment between 30 and 50µm from the soma, p < 0.05 at 110µm from the soma, p < 0.05, 0.01, 0.001 for each 10µm increment between 130 and 150µm from the soma | | | |
| | WT: oGCL vs Contralateral | Fisher's LSD | p < 0.05, 0.05 at 110 and 120µm from the soma, p < 0.05, 0.05 at 190 and 200µm from the soma | | | |
| | WT: iGCL vs oGCL | Fisher's LSD | p < 0.01, 0.01, 0. 01, 0.05 for each 10µm increment between 30µm and 60µm from the soma, p < 0.01, 0.001, 0.001, 0.001, 0.01, 0.001, 0.01, 0.001, 0.01, 0.01 for each 10µm increment between 100 and 190µm from the soma | | | |
| | WT vs ApoE KO: iGCL | Two-way ANOVA | | | | |
| | | Interaction | <0.0001 | 28 | | F (28, 3944) = 4.607 |
| | | Row Factor | <0.0001 | 28 | | F (28, 3944) = 169.7 |
| | | Column Factor | <0.0001 | 1 | | F (1, 3944) = 62.34 |
| | | Fisher's LSD | p < 0.0001; p < 0.05, 0.001, 0.01, 0.001, 0.001, 0.001, 0.001, 0.001, 0.001 and 0.05 for each 10µm increment between 70µm and 170µm from the soma | | | |
| | WT vs ApoE KO: oGCL | Two-way ANOVA | | | | |
| | | Interaction | 0.2707 | 28 | | F (28, 3335) = 1.147 |
| | | Row Factor | <0.0001 | 28 | | F (28, 3335) = 122.3 |
| | | Column Factor | <0.0001 | 1 | | F (1, 3335) = 71.14 |
| | | Fisher's LSD | p < 0.01, 0.01, 0.001, 0.05, 0.01, 0.01, 0.05, 0.05, non-significant, 0.05, 0.05, 0.05 for each 10µm increment between 80µm and 200µm from the soma | | | |
| | WT vs ApoE KO: Contralateral | Two-way ANOVA | | | | |
| | | Interaction | <0.0001 | 28 | | F (28, 4263) = 3.014 |
| | | Row Factor | <0.0001 | 28 | | F (28, 4263) = 203.8 |
| | | Column Factor | <0.0001 | 1 | | F (1, 4263) = 74.25 |
| | | Fisher's LSD | p < 0.001; p < 0.05, non-significant, 0.05, 0.05, 0.05, 0.01, 0.001, 0.001, 0.001, 0.001, 0.001, 0.001 and 0.01 for each 10µm increment between 40µm and 160µm from the soma | | | |
| | ApoE3 vs ApoE4: iGCL | Two-way ANOVA | | | | |
| | | Interaction | <0.0001 | 27 | | F (27, 5068) = 6.679 |
| | | Row Factor | <0.0001 | 27 | | F (27, 5068) = 208.4 |
| | | Column Factor | <0.0001 | 1 | | F (1, 5068) = 143.9 |
| | | Fisher's LSD | p < 0.01, 0.001, 0.001, 0.001, 0.001, 0.001, 0.001, 0.001, 0.001, 0.001, 0.001, 0.001, 0.001 and 0.05 for each 10µm increment between 50µm and 180µm from the soma | | | |
| | ApoE3 vs ApoE4: oGCL | Two-way ANOVA | | | | |
| | | Interaction | <0.0001 | 27 | | F (27, 3500) = 3.279 |
| | | Row Factor | <0.0001 | 27 | | F (27, 3500) = 162.6 |
| | | Column Factor | <0.0001 | 1 | | F (1, 3500) = 151.8 |
| | | Fisher's LSD | p < 0.05, 0.001, 0.001, 0.001, 0.001, 0.001, 0.01, 0.01, 0.001, 0.001, 0.001, 0.001, 0.001, 0.01 and 0.001 for each 10µm increment between 50µm and 190µm from the soma | | | |
| | ApoE3 vs ApoE4: Contralateral | Two-way ANOVA | | | | |
| | | Interaction | <0.0001 | 27 | | F (27, 4704) = 4.133 |
| | | Row Factor | <0.0001 | 27 | | F (27, 4704) = 217.8 |
| | | Column Factor | 0.0419 | 1 | | F (1, 4704) = 4.143 |
| | | Fisher's LSD | p < 0.01, 0.001, 0.001 and 0 0.001 for each 10µm increment between 40µm and 70µm from the soma, p < 0.05 at 100µm from the soma, p < 0.01, 0.01, 0.01 and 0.01 for each 10µm increment between 180µm and 210µm from the soma | | | |

DF = Degrees of Freedom; iGCL = Inner Granule Cell Layer, oGCL = Outer Granule Cell Layer, in the ipsilateral dentate gyrus; WT = Wild Type. KO = Knockout. The Sholl Analysis represents the average number of intersections in each condition (Column factor) function of the distance to the soma (Row factor).

confirmed an absence of ApoE staining in the ApoE null animal (Fig 1G–1K). Finally, we confirmed ApoE astrocytic staining in the targeted replacement ApoE3 (Fig 1L–1P) and ApoE4 (Fig 1Q–1U) mouse dentate gyrus, indicating close physical interaction of astrocytes with maturing injury-induced neurons (Fig 1P–1U, white arrowheads).

## Injury-induced adult-born neurons lacking ApoE show impaired dendritic development after CCI

Since it is known that ApoE deficiency leads to less sophisticated granule neurons during postnatal neurogenesis [42] and that CCI induces misplacement of adult-born granule neurons in

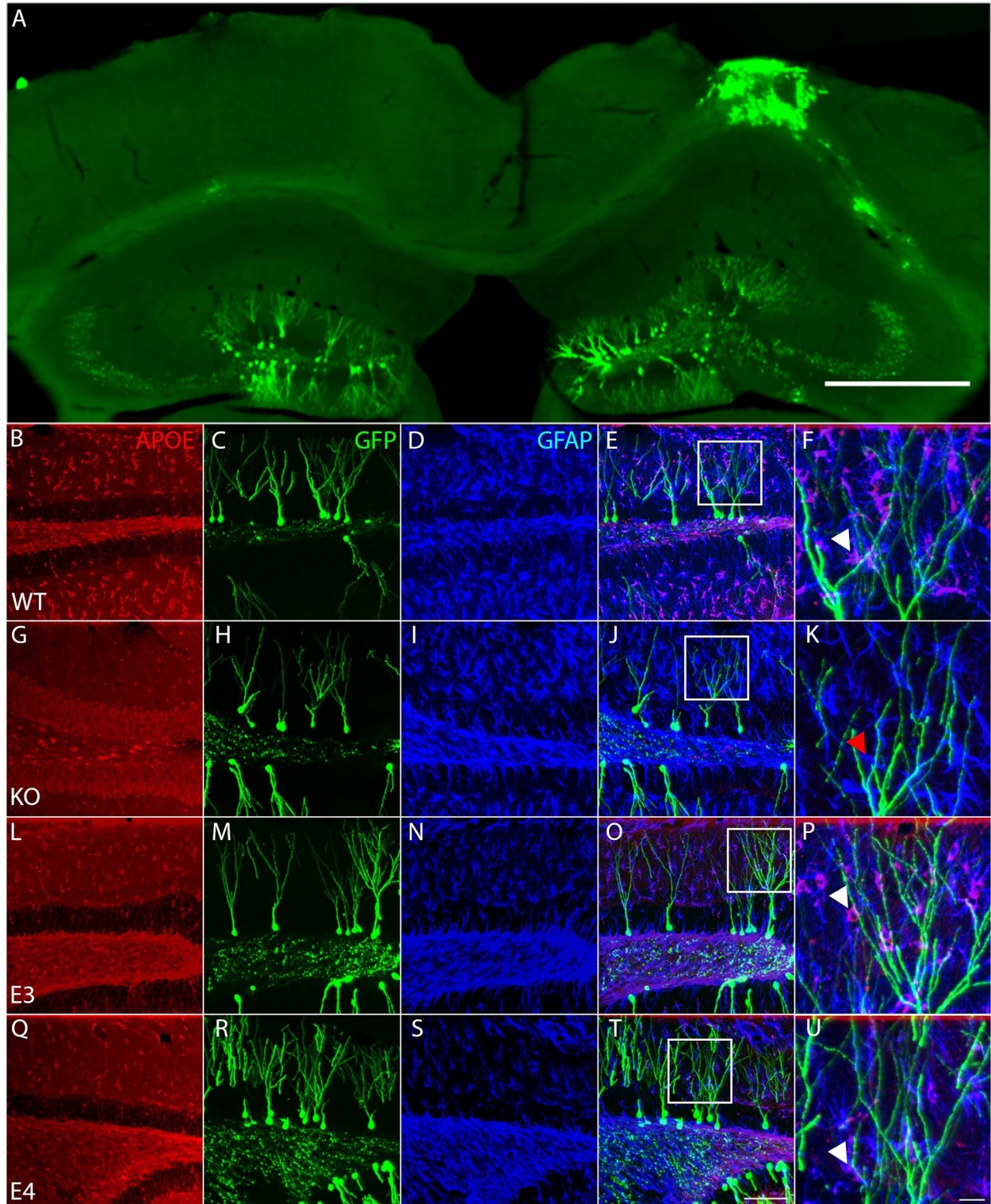

**Fig 1. ApoE-expressing astrocytes physically interact with injury-induced GFP-expressing dendrites.** (**A**) Representative section of the contralateral and ipsilateral cortex and hippocampus 4 weeks after Controlled Cortical Impact and stereotaxic injection of GFP-expressing retroviruses in the dentate gyrus of a wildtype mouse reveals the typical morphology of the injured hippocampus along with efficient long-term labelling of neural progenitor cells. Scale bar 500μm. (**B-U**) Representative pictures of ApoE (first column, antigen is recombinant human ApoE and can react with human e2, e3 and e4, as well as non-human primate and mouse ApoE), GFP (second column) and GFAP (third column)

immunostaining in the dentate gyrus of WT (**B-F**), ApoE KO (**G-K**), ApoE3 (**L-P**) and ApoE4 mice (**Q-U**). (**E, J, O, T**) Merged pictures (scale bar 100μm), white squares indicate enlarged area shown in last column (**F, K, P, U**), white arrowheads designate ApoE-expressing GFAP-positive astrocytes wrapping around GFP-expressing dendrites, red arrowhead designate ApoE-deficient GFAP-positive astrocyte, scale bar 10μm.

the dentate gyrus, we investigated the effect of ApoE deficiency on the dendritic maturation of CCI-induced newborn neurons.

We observed no differences in the distance to the first branch when comparing WT to ApoE-deficient injury-induced neurons whether they were in the inner or outer portions of the ipsilateral dentate gyrus or on the contralateral side (Fig 2A). However, when we compared the number of nodes and the total dendritic length, we observed highly significant attenuation in the ApoE-deficient dentate gyrus (Fig 2B–2D). We also observed significantly narrower dendritic trees in the ApoE-deficient neurons found in the contralateral dentate gyrus, when compared to WT matching conditions (Fig 2E), representative traces are shown in Fig 2C.

We next studied the distribution of the dendritic branches with a Sholl Analysis. A two-way ANOVA was conducted to quantify the effect of the different conditions on the dendritic complexity relative to the distance to the soma. We found a significant interaction in the iGCL and significant column (number of intersections) and row (distance to the soma) effects in both the iGCL and oGCL. Fisher's LSD post-hoc test exposed an attenuation of branching number in ApoE KO adult-born granule neurons found in the iGCL when compared to WT (Fig 2F), as well as in the oGCL (Fig 2G) and in the contralateral GCL (Fig 2H).

As previously described [42], imaging of adult-born granule cells revealed that their distribution and number of dendritic branches were not uniform independent of genotype or soma localization. To quantify this observation, for each condition, we categorized neuronal traces as a function of their dendritic complexity 4 weeks following GFP-expressing retroviral infection (Fig 2I–2K). Consistent with previous findings [42], we observed that adult-born granule cells lacking ApoE demonstrated a greater number of cells with up to 4 nodes, particularly in the oGCL (23.3%) when compared with WT (6.4%; Fig 2J), while we detected that more than 40% of WT injury-induced newborn neurons dendritic arborizations had 8 nodes or more compared to ~23% for ApoE-deficient mice in the iGCL (Fig 2I) and contralateral side (Fig 2J), and ~25% in the oGCL (Fig 2K). This result highlights that the attenuation in dendritic complexity in ApoE-deficient adult-born neurons following CCI is due to the presence of a higher proportion of less complex neurons, particularly in the oGCL. Finally, we examined the spine density of dentate gyrus adult-born neurons lacking ApoE, by reconstructing randomly selected 10μm dendritic fragments from GFP-expressing cells. The reduced spine density observed in ApoE-deficient adult-born granule neurons [42] is also present after CCI (Fig 2L and 2M).

Together, these results demonstrate morphological dendritic aberrancies of injury-induced neurons in the ApoE deficient mouse dentate gyrus compared to WT mice in all three studied areas.

## ApoE4 leads to less complex adult-born granule neurons after CCI when compared to ApoE3

We next compared the development of adult-born granule neurons in human ApoE4 and ApoE3 mice following controlled cortical impact injury. We examined the first branch length and found a significant difference based on genotype and further analysis revealed differences for neurons located in the iGCL and the contralateral side but not the oGCL (Fig 3A). We then compared the complexity of injury-induced neurons in the different conditions as a function of genotype. This analysis revealed a significant effect on complexity and post-hoc tests

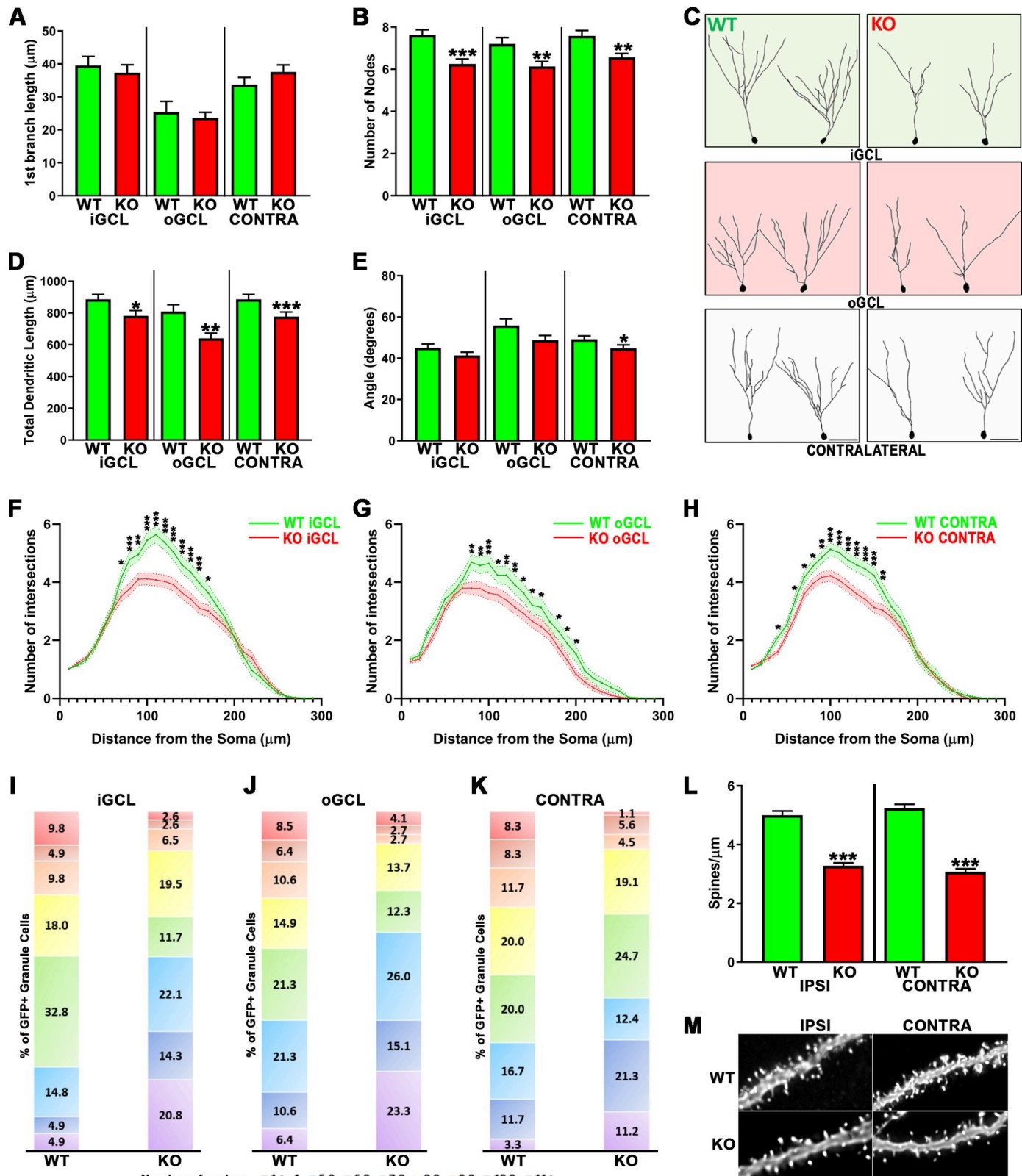

**Fig 2. ApoE deficiency leads to impaired dendritic development of injury-induced adult-born granule neurons.** (**A**) ApoE deficiency did not affect the distance to the first dendritic branching but did significantly affect dendritic complexity (**B**) in the iGCL (p < 0.001), oGCL (p < 0.01) and contralateral (p < 0.01) dentate gyrus when compared to WT. (**C**) Representative tracings of GFP-expressing cells found in the iGCL, oGCL and contralateral dentate gyrus from WT and ApoE-

deficient injured dentate gyrus. (**D**) The cumulative dendritic length is significantly attenuated in mouse lacking ApoE when compared to matching Wild Type cells (iGCL: $p < 0.05$, oGCL: $p < 0.01$, Contra: $p < 0.001$). (**E**) Injury-induced cells found in the contralateral dentate gyrus in the absence of ApoE also showed decreased dendritic span when compared to WT ($p < 0.05$). Sholl analysis of dendritic arborizations of CCI-induced adult-born granule neurons exposed differences in the dendritic branching number observed in both proximal and distal regions when comparing WT with ApoE-deficient GFP-expressing cells found in the iGCL (**F**), in the oGCL (**G**) and in the contralateral dentate gyrus (**H**). (**I-K**) In each condition, all traced neurons have been categorized depending on their complexity ($\leq$4...$\geq$11 nodes), highlighting higher proportions of less complex neurons (<4 nodes) in ApoE-deficient, particularly in the oGCL. 4 mice/condition and at least 10 neurons/mouse were analyzed; WT iGCL: 61 cells; WT oGCL: 47 cells; WT Contra: 60 cells; ApoE KO iGCL: 77 cells; ApoE KO oGCL: 73 cells; ApoE KO Contra: 89 cells. Contra = Contralateral dentate gyrus; iGCL = Inner Granule Cell Layer, oGCL = Outer Granule Cell Layer, in the ipsilateral dentate gyrus; WT = Wild Type, KO = ApoE Knock-out. ***$p < 0.001$. (**L**) ApoE deficiency leads to significantly reduced spine density in injury-induced adult-born granule neurons when compared to WT cells from both the Ipsilateral and Contralateral sides ($p < 0.001$). High power representative pictures of dendritic fragments from mature adult-born granule neurons in wildtype and ApoE-deficient (**M**). 4 mice/condition; Number of dendritic fragments analyzed: WT Ipsilateral = 60, WT Contralateral = 62, ApoE KO Ipsilateral = 87, ApoE KO Contralateral = 53. Scale bar = 5μm.

indicated that in the iGCL, oGCL, and contralateral side to the injury, ApoE4 adult-born neurons have fewer nodes when compared to ApoE3 (Fig 3B), as well as having reduced cumulative dendritic length in both the iGCL and oGCL (Fig 3D). Representative traces are shown in Fig 3C.

We next examined the dendritic span of adult-born GC dendritic arborizations by measuring the angle formed by the two most distal branches when projected onto two dimensions. CCI had a significant effect on ApoE4 dendritic span as shown by Kruskal-Wallis test and Dunn's post-hoc tests for cells located in the iGCL and contralateral dentate gyrus but not in the oGCL (Fig 3E). Finally, we studied the distribution of the dendritic branches with a Sholl Analysis. A two-way ANOVA was conducted that examined the effect of the different conditions on the number of dendritic intersections relative to the distance from the soma. We found a significant interaction of the parameters in the Inner GCL (Fig 3F), as well as in the oGCL (Fig 3G) and in the contralateral side GCL (Fig 3H).

We then categorized neuronal traces as a function of their dendritic complexity and demonstrated that 4 weeks following eGFP-infusions. Consistent with previous findings [42], we observed that adult-born granule cells expressing the E4 human isoform demonstrated a greater number of cells with up to 4 nodes, particularly in the oGCL (24.6%) compared with ApoE3 (6.5%; Fig 3J), while we detected that ~50% of ApoE3 injury-induced newborn neurons dendritic arborizations had 8 nodes or more compared to ~30% for ApoE4 mice in the iGCL (Fig 3I) and contralateral side (Fig 3J), and ~25% in the oGCL (Fig 3K). This result highlights that similarly to what we observed with ApoE deficient mice, the attenuation in dendritic complexity in ApoE4 adult-born neurons following CCI is due to the presence of a higher proportion of less complex neurons, particularly in the oGCL. Finally we studied the spine density of dentate gyrus adult-born neurons expressing the ApoE4 human isoform by reconstructing randomly selected 10μm dendritic fragments from GFP-expressing cells. The reduced spine density observed in ApoE4 adult-born granule neurons (29) is also present after CCI (Fig 3L and 3M).

Together, these results demonstrate morphological dendritic aberrancies of injury-induced neurons in the ApoE4 mouse dentate gyrus compared to ApoE3 mice in all three studied areas.

Finally, we describe the intra-condition comparisons of dendritic morphology to highlight the changes driven by the injury in each genotype. Both the dendritic complexity and the spine density of injury-induced adult-born granule neurons were similar in the 3 studied areas (iGCL/oGCL and Contralateral hemispheres) for all 4 genotypes, revealing no injury-driven effect for these parameters. Thus, the differences observed in dendritic morphology and spine density when comparing WT with KO or E3 with E4 were strictly driven by genotype (S1 and S2 Figs).

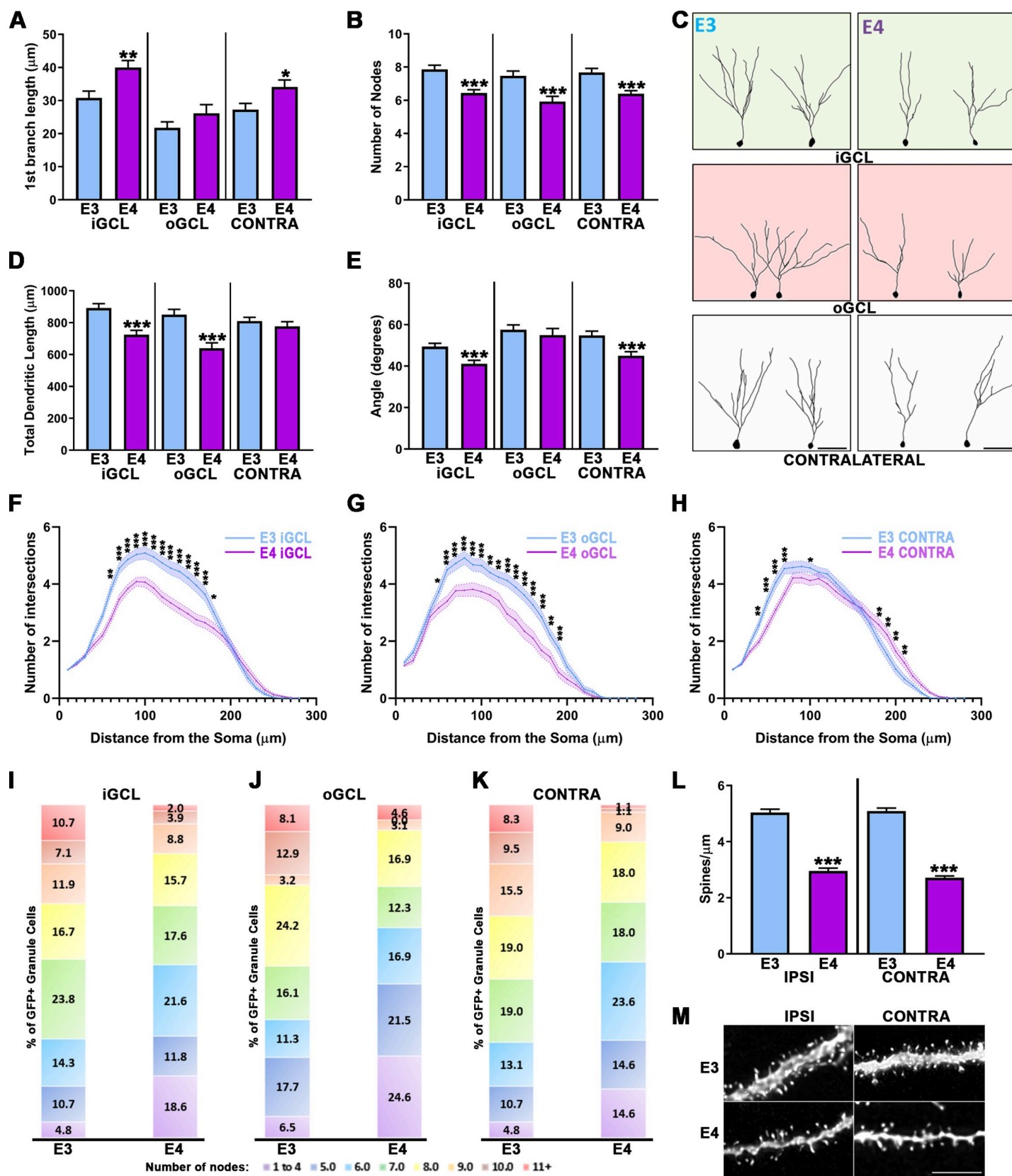

**Fig 3. Impaired dendritic development of adult-born granule neurons after CCI in ApoE4 mouse. (A)** GFP-cells found in the iGCL (p < 0.01) and contralateral dentate gyrus (p < 0.05) of ApoE4 mice have the first dendritic branching occurring further from their cell body when compared to ApoE3. (**B**) The number of dendritic intersections of adult-born granule cells located in the iGCL, oGCL and contralateral dentate gyrus of ApoE4 mice is significantly reduced when compared to ApoE3 (p < 0.001). (**C**) Representative tracings of GFP-expressing cells found in the iGCL, oGCL and contralateral dentate gyrus from ApoE3 and ApoE4 adult mouse injured dentate gyrus. (**D**) The cumulative length of ApoE4-expressing injury-induced neurons' dendrites is significantly reduced in both the iGCL and oGCL (p < 0.001). (**E**) ApoE4 significantly impairs the dendritic span of adult-born granule cells from both the iGCL and the contralateral dentate gyrus (p < 0.001). Sholl analysis revealed significant differences in the number of dendritic intersections observed in both proximal and distal regions when comparing ApoE3 with ApoE4-expressing cells found in the iGCL (**F**), in the oGCL (**G**) and in the contralateral dentate gyrus (**H**). 4 mice/condition and at least 10 neurons/mouse were analyzed; ApoE3 iGCL: 81 cells; ApoE3 oGCL: 62 cells; ApoE3 Contra: 81 cells; ApoE4 iGCL: 102 cells; ApoE4 oGCL: 65 cells; ApoE4 Contra: 89 cells. Contra = Contralateral dentate gyrus; iGCL = Inner Granule Cell Layer, oGCL = Outer Granule Cell Layer, in the ipsilateral dentate gyrus. *p < 0.05, **p < 0.01, ***p < 0.001. (**I-K**) In each condition, all traced neurons have been categorized depending on their complexity (≤4...≥11 nodes), highlighting higher proportions of less complex neurons (<4 nodes) in ApoE4-expressing mice, particularly in the oGCL. 4 mice/condition and at least 10 neurons/mouse were analyzed; ApoE3 iGCL: 81 cells; ApoE3 oGCL: 62 cells; ApoE3 Contra: 84 cells; ApoE4 iGCL: 102 cells; ApoE4 oGCL: 65 cells; ApoE4 Contra: 82 cells. Contra = Contralateral dentate gyrus; iGCL = Inner Granule Cell Layer, oGCL = Outer Granule Cell Layer, in the ipsilateral dentate gyrus; E3 = ApoE3, E4 = ApoE4. ***p < 0.001. (**L**) ApoE4-expressing adult-born granule neurons show significantly attenuated spine density when compared to ApoE3 matching cells from both dentate gyrus (p < 0.001). High power representative pictures of dendritic fragments from mature adult-born granule neurons in ApoE3 and ApoE4 (**M**) ipsilateral and contralateral dentate gyrus. 4 mice/condition; Number of dendritic fragments analyzed: ApoE3 Ipsilateral = 99, ApoE3 Contralateral = 91, ApoE4 Ipsilateral = 71, ApoE4 Contralateral = 58. Scale bar = 5μm.

## Discussion

Previous studies revealed that ApoE deficiency in mice leads to a depletion of the neural stem and progenitor pool over time with a subsequent decrease in the activation of Type I cells. This observation is accompanied by further impaired dendritic development in mice lacking ApoE or expressing the human isoform ApoE4 [42]. It is also known that ApoE is necessary for normal injury-induced neurogenesis following experimental moderate TBI, and is associated with ApoE human isoform-dependent effects [17]. In the present study, we establish that ApoE deficiency and human ApoE genotype both influence the dendritic development of newborn granule neurons in the injured adult mouse hippocampus, which includes impaired dendritic arborization and diminished spine density in mice either lacking ApoE or expressing human ApoE4. We also observe that CCI induces outward migration of granule neurons into the oGCL, which is why we further divided adult-born granule neurons from the ipsilateral side into the inner one-third or outer two-thirds of the GCL. Importantly, and contrary to what has been previously shown [7], CCI did not affect the complexity, total dendritic length, or spine density of mature adult-born granule neurons, 4 weeks after brain injury (when compared to contralateral matching neurons). These findings are consistent with a recent study in which experimental ischemic stroke was induced in adult mice [48]. Finally, by grouping adult-born granule neurons in the different conditions as a function of their complexity, we further highlighted that the observed attenuation in dendritic branching of injury-induced granule cells lacking ApoE or expressing the ApoE4 human isoform was the result of a higher proportion of less complex neurons when compared to WT and ApoE3.

TBI triggers selective secondary cell death in the dentate gyrus and immature granule neurons localized in the inner GCL constitute the most vulnerable population while both nestin-expressing NSPCs and NeuN-expressing mature granule cells have been shown to be less vulnerable [4, 5]. TBI commonly causes cognitive functional deficits though some degree of spontaneous recovery occurs, which appears to be mediated at least in part by injury-induced hippocampal neurogenesis [3]. It has also been shown that injury-induced adult-born granule neurons mature more rapidly than those observed in usual adult neurogenesis [7, 48]. These observations suggest that injury-induced neurogenesis may in part compensate for the loss of young adult-born granule neurons by generating new adult-born granule neurons and accelerating their dendritic development.

A typical hallmark of TBI is the outward migration of injury-induced neurons in the ipsilateral dentate gyrus to the outer two-thirds of the granule cell layer at one week and one month

after CCI [10]. Here, we observe that, after CCI, most new neurons were found in the oGCL, where their first dendritic branching was found closer to the cell body than what is observed in the iGCL. In addition, these newborn neurons demonstrated a greater lateral spread of their dendrites, suggesting that their morphology is not aberrant, but adaptive. This is supported by the fact that new neurons found in the oGCL are not less sophisticated, though they do show some degree of dendritic reorganization. Moreover, injury-induced neurons do not develop aberrant electrophysiological properties as they demonstrate preserved early electrophysiological maturation and functionally integrate to the pre-existing try-synaptic circuitry four weeks following CCI [7], supporting the idea that they can function normally.

The key regulators of neuronal migration (i.e. onset, speed, directionality, and arrest) remain poorly understood. It has been shown that experimental stroke [49], neonatal hypoxia [50], entorhinal cortical lesions [51], and seizures [8, 15, 52, 53] all induce ectopic migration of newborn granule cells and/or dendritic arborization aberrancies. Several mechanisms have been implicated in the outward migration of injury-induced adult-born granule neurons including cell-intrinsic impairments, regionally localized abnormalities, granule cell layer dispersion, and somatic translocation [8]. It is unclear why approximately half of the adult-born granule neurons migrate to the oGCL after CCI, while the NSPCs may be functionally normal, the injury-induced environment may disrupt migratory cues, which would lead to ectopic migration. This is supported by recent work *ex vivo* which suggests that the regionalized presence of the extracellular matrix protein reelin in the molecular layer controls the directionality of granule cell migration, but not the actual migratory process or speed [54].

## ApoE genotype influences dendritogenesis

We demonstrate here that the previously observed morphologic impairments of adult-born granule neurons lacking ApoE or expressing the human ApoE4 isoform also occur after experimental TBI, but in a much more exaggerated manner. Indeed, after CCI, adult-born granule cells from mice lacking ApoE or expressing ApoE4 show impaired dendritic arborization (i.e. complexity, cumulative length, Sholl analysis, and spine density). Moreover, both the dendritic complexity and the spine density of injury-induced adult-born granule neurons were similar in the 3 studied areas (iGCL/oGCL and Contralateral hemispheres) for all 4 genotypes, revealing no injury-driven effect for these parameters (S1 & S2 Figs). Thus, the differences observed in dendritic morphology and spine density when comparing WT with KO or E3 with E4 were strictly driven by genotype. Interestingly, we observe that ApoE deficiency nearly phenocopies the substitution of mouse ApoE with human ApoE4, thereby strengthening the hypothesis that ApoE4 works more as negative regulator of hippocampal neurogenesis and development. This is consistent with our previous published observations regarding ApoE and neurogenesis but the underlying mechanisms remain unclear [17, 21, 42].

ApoE is the major lipid carrier of the brain and participates primarily in cholesterol transportation from astrocytes to neurons, where it affects both dendritic development and synaptogenesis in an LDL receptor-dependent manner [55]. The role of cholesterol in neurite development and synapse formation were established first *in vitro* [56, 57] where it has been shown that while human ApoE3 promotes neurite outgrowth, ApoE4 inhibits it [58]. Since granule neurons do not express ApoE once they mature [20], the observed dendritic impairments are likely from astrocytic-ApoE, which is consistent with several studies that demonstrated an astrocytic requirement for normal dendritic development [59–61]. More recently, ApoE isoform-dependent roles in regulating synaptic pruning by astrocytes in the developing mouse brain have been established [62].

Together, these previous findings may help to explain the cognitive impairments exhibited by mice lacking ApoE or expressing ApoE4 particularly in spatial learning/memory and in olfactory memory, two neurogenic-dependent behaviors [63–71]. Although it remains unclear whether recovery from injuries such as TBI is dependent on neurogenesis in general and ApoE state in particular, the present study adds needed insight into a potential mechanism linking the two.

## Conclusions

In the present study, we demonstrate that the dendritic complexity, dendritic length, and spine density of injury-induced mature adult-born granule neurons are not impaired 4 weeks after CCI in WT adult mice. We have, however, observed dendritic reorganization in injury-induced adult-born granule neurons, suggesting that they adapt their dendritic tree in response to the locally injured environment. We have also further uncovered the central role for ApoE in injury-induced neurogenesis in the adult hippocampus, where it influences dendritogenesis and synaptogenesis. Because injury-induced neurogenesis is necessary for at least some aspects of cognitive recovery, the dendritic impairments observed in mice lacking ApoE or expressing the ApoE4 human isoform may help inform the greater risk for poor outcomes in individuals possessing an ApoE4 genotype and subjected to TBI, and strengthen the ApoE4 allele association with the emergence of hippocampal-related cognitive decline.

## Supporting information

**S1 Fig. WT and KO intra-condition comparisons of dendritic morphology and Sholl analysis.** WT injury-induced adult-born granule neurons found in the oGCL branched closer to their cell body when compared to adult-born granule neurons found in the iGCL (p < 0.001) or contralateral (p < 0.01) dentate gyrus (**A**) while the number of dendritic divisions is similar between adult-born granule neurons independently of their soma localization (**B**). (**C**) Sholl analysis exposed significant differences in dendritic branching observed in both proximal and distal regions when comparing the oGCL with the Contralateral side (red stars) and in more distal regions when comparing iGCL with Contralateral side (green stars). (**D**) The cumulative dendritic length is similar in the three conditions. (**E**) Injury-induced adult-born granule neurons found in the oGCL have a wider dendritic span when compared to matching cells from the contralateral side (p < 0.05) but not the iGCL. (**F**) Sholl analysis also revealed differences in the dendritic patterns of cells found in the iGCL with the oGCL in both proximal and distal regions. 4 mice/condition and at least 10 neurons/mouse were analyzed; iGCL: 61 cells; oGCL: 45 cells; Contra: 60 cells. ApoE KO injury-induced adult-born granule neurons found in the oGCL branched closer to their cell body when compared to adult-born granule neurons found in the iGCL (p < 0.001) or contralateral (p < 0.001) dentate gyrus (**G**) while the number of dendritic divisions is similar between adult-born granule neurons independently of their soma localization (**H**). (**I**) Sholl analysis exposed significant differences in dendritic branching observed in both proximal and distal regions when comparing the oGCL with the Contralateral side (red stars) and in more distal regions when comparing iGCL with Contralateral side (green stars). (**J**) The cumulative dendritic length was decreased in neurons found in the oGCL when compared to iGCL or Contralateral side (p < 0.001). (**K**) Injury-induced adult-born granule neurons found in the oGCL have a wider dendritic span when compared to matching cells from the iGCL (p < 0.05) but not the contralateral side. (**L**) Sholl analysis also revealed differences in the dendritic patterns of cells found in the iGCL with the oGCL in both proximal and distal regions. 4 mice/condition and at least 10 neurons/mouse were analyzed; iGCL: 77 cells; oGCL: 73 cells; Contra: 89 cells; iGCL = Inner Granule Cell Layer,

oGCL = Outer Granule Cell Layer, in the ipsilateral dentate gyrus. Contra = Contralateral dentate gyrus. *p < 0.05, **p < 0.01, ***p < 0.001. Detailed statistics can be found in S1 Table. (TIF)

**S2 Fig. ApoE3 and ApoE4 intra-condition comparisons of dendritic morphology and Sholl analysis.** ApoE3 injury-induced adult-born granule neurons found in the oGCL branched closer to their cell body when compared to adult-born granule neurons found in the iGCL (p < 0.01) or contralateral (p < 0.05) dentate gyrus (**A**) while the number of dendritic divisions is similar between adult-born granule neurons independently of their soma localization (**B**). (**C**) Sholl analysis exposed significant differences in dendritic branching observed in both proximal and distal regions when comparing the iGCL with the contralateral side (blue stars) and minor differences in proximal regions when comparing oGCL with contralateral side (purple stars). (**D**) The cumulative dendritic length in the contralateral side was reduced when compared to iGCL (p < 0.05). (**E**) Injury-induced adult-born granule neurons found in the oGCL have a wider dendritic span when compared to matching cells from the iGCL (p < 0.05) but not the contralateral side. (**F**) Sholl analysis also revealed differences in the dendritic patterns of cells found in the iGCL with the oGCL in both proximal and distal regions. 4 mice/ condition and at least 10 neurons/mouse were analyzed; iGCL: 81 cells; oGCL: 62 cells; Contra: 81 cells. ApoE4 injury-induced adult-born granule neurons found in the oGCL branched closer to their cell body when compared to adult-born granule neurons found in the iGCL (p < 0.001) or contralateral (p < 0.01) dentate gyrus (**G**) while the number of dendritic divisions is similar between adult-born granule neurons independently of their soma localization (**H**). (**I**) Sholl analysis exposed significant differences in dendritic branching observed in both proximal and distal regions when comparing the oGCL with the Contralateral side (purple stars) while minor differences have been found when comparing iGCL with Contralateral side (blue stars). (**J**) The cumulative dendritic length was decreased in neurons found in the oGCL when compared to iGCL (p < 0.05) or contralateral side (p < 0.01). (**K**) Injury-induced adult-born granule neurons found in the oGCL have a wider dendritic span when compared to matching cells from the iGCL (p < 0.001) and the contralateral side (p < 0.05). (**L**) Sholl analysis also revealed differences in the dendritic patterns of cells found in the iGCL with the oGCL in both proximal and distal regions. 4 mice/condition and at least 10 neurons/mouse were analyzed; iGCL: 102 cells; oGCL: 65 cells; Contra: 89 cells; iGCL = Inner Granule Cell Layer, oGCL = Outer Granule Cell Layer, in the ipsilateral dentate gyrus. Contra = Contralateral dentate gyrus. *p < 0.05, **p < 0.01, ***p < 0.001. Detailed statistics can be found in S1 Table. (TIF)

**S1 Table. Summary of statistics for intra-genotype comparisons.** DF = Degrees of Freedom; iGCL = Inner Granule Cell Layer, oGCL = Outer Granule Cell Layer, in the ipsilateral dentate gyrus; WT = Wild Type. KO = Knockout. The Sholl Analysis represents the average number of intersections in each condition (Column) function of the distance to the soma (Row). (PDF)

## Acknowledgments

Moloney Murine Leukemia Viral vectors were generated by the GT3 Core Facility of the Salk Institute (La Jolla, CA).

## Author Contributions

**Conceptualization:** Tzong-Shiue Yu, Steven G. Kernie.

**Methodology:** Yacine Tensaouti.

**Supervision:** Tzong-Shiue Yu, Steven G. Kernie.

**Writing – original draft:** Yacine Tensaouti.

**Writing – review & editing:** Steven G. Kernie.

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
