## [Decision Letter · Decision Letter 0]

29 Oct 2019

PONE-D-19-27231

Apolipoprotein E regulates the maturation of injury-induced adult-born hippocampal neurons following traumatic brain injury

PLOS ONE

Dear Dr. Kernie,

Thank you for submitting your manuscript to PLOS ONE. After careful consideration by 2 Reviewers and an Academic Editor, all of the critiques of both Reviewers must be addressed in detail in a revision to determine publication status. If you are prepared to undertake the work required, I would be pleased to reconsider my decision, but revision of the original submission without directly addressing the critiques of the two Reviewers does not guarantee acceptance for publication in PLOS ONE. If the authors do not feel that the queries can be addressed, please consider submitting to another publication medium. A revised submission will be sent out for re-review. The authors are urged to have the manuscript given a hard copyedit for syntax and grammar.

**Comments to the Author**

1. Is the manuscript technically sound, and do the data support the conclusions?

Reviewer #1: Yes

Reviewer #2: Partly

2. Has the statistical analysis been performed appropriately and rigorously? 

Reviewer #1: Yes

Reviewer #2: Yes

3. Have the authors made all data underlying the findings in their manuscript fully available?

Reviewer #1: Yes

Reviewer #2: Yes

4. Is the manuscript presented in an intelligible fashion and written in standard English?

Reviewer #1: Yes

Reviewer #2: Yes

5. Review Comments to the Author

Reviewer #1: This manuscript describes the influence of APOE genotype on morphological characteristics of adult born granule cells. They have used stereotactic injections of GFP expressing retroviruses which labels adult born hippocampal neurons. They have also used the stereoinvestigator and neurolucida platforms to perform their analyses of morphological changes to dendrites and spines of adult born granule cells.

The findings from this manuscript supports previous studies in the field and provide some new perspectives in the morphological changes in the brains of injured APOE-KO and APOE4 mice. However the reviewer feels that separate analyses and a reanalyses of their datasets are needed, including a reconsideration of their conclusions.

Introduction:

46-52: If TBI activates NSPCs to give rise to adult born granule neurons, and ablation of injury induced hippocampal adult born neurons impairs cognitive recovery after TBI, does this imply that the injury effect is neuroprotective? Would be helpful if the "controversy" or different ideas are explained a little further to lay the groundwork as it regards to the general consensus in the field.

Types of TBI are not addressed (in relation to APOE/TBI outcomes and TBI/neurogenesis effects).

Effect of APOE4 in the context of TBI involves different mechanisms, which have been well explored molecularly in several studies. How much neurogenesis plays in the APOE4 dependent neurodegenerative or behavioral effects remains unknown?

Significant papers downplaying the role of neurogenesis in TBI are not referenced.

Design/Methods/results:

No sham mice have been included, as the contralateral hemisphere is used as a control. This is problematic and needs to be addressed.

As a result the authors focus on genotype effects in injured mice alone.

The authors need to include sham mice (sham injuries/cranial window and injections only), or at least demonstrate in this manuscript that sham mice (ipsilateral hemisphere with cranial window) show comparable changes with the contralateral hemisphere of injured mouse brains.

Additionally, data needs to be presented to show that there were no difference in the iGCL vs oGCL of the contralateral region.

There is no mention of injury effects within each genotype for APOE KO, APOE3 and APOE4 mice (Figure 2 and 3, 4A-B). For example no changes in spine density (between ipsi vs contra) was observed in WT, KO, E3 and E4 mice. In this case, the changes in the contra hemisphere were similar to the changes in the ipsi hemisphere, showing that the effects are driven by genotype and not injury. This should be highlighted and discussed.

Data in Figure 1 are included in graphs/analyses of data in Figure 2 and 4 which is redundant and repetitive.

It will be better to include the analyses of all groups on the same graph as in Fig 4E-G (This would be very busy but easier for the reader to compare across the different groups if clear labeling is used).

Rationale for including APOE-KO mice is not clear, as this is not relevant to humans (the main question is the influence of APOE3 vs APOE4). Moreover comparison of APOE-KO mice with WT also does not add much to our understanding as mice express a different form of APOE.

Why did the authors not collect cell counts in their analyses using the optical fractionator?

It was stated that astrocytes drive NSPCs to astrogenesis, it would add value to the paper if the authors obtained double staining for differentiated astrocytes as they did in their previous paper with nestin, ki-67 and GFAP etc... and APOE (Tensaouti et al., 2018).

If mice were administered Brdu, it would be good to include Brdu/Prox1 cell density in the different granular layers.

200: Says "mixed genotype and....." however the legend states that the data were from WT mice.

What was the rationale for injuries at 6 weeks, this is still very young and the brain is still developing in mice.

Discuss in some more detail, the rationale for the injection at 4 weeks post-injury. How does this relate to the timeline of events after CCI, and what could be missed from this timepoint of analyses.

211: is there another term that could be used instead of "spread" of adult granule neurons.

Discussion:

Their discussion needs to be expanded and re-written based on the comments above.

Reviewer #2: In the manuscript, the authors investigated the effects of ApoE gene in the regulation of the maturation of injury-induced adult-born neurons in the hippocampus following traumatic brain injury. Overall, the manuscript is interesting and contributes to the understanding of ApoE genes in the regulation of adult hippocampal neurogenesis after brain injury. However, some concerns need to be addressed.

1. In the study, mixed sex mice were used as the authors claimed. What is the percentage of each sex mice in each group? Are there the same numbers of female and male mice investigated in each genotype? It is known that there is sex difference in basal adult neurogenesis, however, the authors did not mention the numbers of male and female mice in each group they studied.

2. The authors should provide more information about the retrovirus carring eGFP they used in the study, it will help the readers understand the work more easily.

3. In result2, the authors claimed that they found significant interaction in both the iGCL and oGCL. However, in table1, the results shows the interaction of sholl analysis in oGCL is not significant.

4. In the two-way ANOVA they used to analyze the sholl anaylasis results, they provided the details of the results. However, what is the row factor in the analysis? They should specifically point out what is row factor and what is column factor.

5. In result4, they claimed that "the reduced spine density in ApoE-KO mice adult-born neurons is exaggerated after CCI". However, the result of the ApoE-KO adult born neurons without CCI is published in another paper. I am wondering if the authors conduct the two studies by using the mice purchased at the same time and performed the studies at the same time? If not, it is not very appropriate to compare these two results.

6. More details about the ApoE-KO, ApoE3 and ApoE4 mice will help the readers understand the manuscript better.

6. PLOS authors have the option to publish the peer review history of their article (what does this mean?). If published, this will include your full peer review and any attached files.

**Do you want your identity to be public for this peer review?** For information about this choice, including consent withdrawal, please see our Privacy Policy.

Reviewer #1: No

Reviewer #2: Yes: Hao Wang

We would appreciate receiving your revised manuscript by February 2020. To enhance the reproducibility of your results, we recommend that if applicable you deposit your laboratory protocols in protocols.io, where a protocol can be assigned its own identifier (DOI) such that it can be cited independently in the future. For instructions see: http://journals.plos.org/plosone/s/submission-guidelines#loc-laboratory-protocols

We look forward to receiving your revised manuscript.

Kind regards,

Stephen D. Ginsberg, Ph.D.

Section Editor

PLOS ONE
---

## [Author Response · Author response to Decision Letter 0]

29 Jan 2020

Reviewer #1: INTRODUCTION

1)46-52: If TBI activates NSPCs to give rise to adult born granule neurons, and ablation of injury induced hippocampal adult born neurons impairs cognitive recovery after TBI, does this imply that the injury effect is neuroprotective? Would be helpful if the "controversy" or different ideas are explained a little further to lay the groundwork as it regards to the general consensus in the field.

We thank the reviewer for highlighting an ongoing controversy regarding injury-induced neurogenesis. It is clear that rodent modeling of TBI using a variety of injury modeling including the controlled cortical impact (CCI) model that we use here leads to neuronal cell death in the dentate with loss of both mature and immature DCX+ expressing neurons (Gao et al., 2008, Yu et al., 2008). It has also been shown that injury-induced activation of resident progenitors leads to neurogenesis that compensates for the loss of these vulnerable cell populations. These injury-induced neurons functionally integrate into the pre-existing circuitry (Villasana et al., 2015) and appear to be necessary for at least some aspects cognitive recovery in mice (Blaiss et al., 2011). In addition, it has been reported that these new neurons have morphologic aberrancies that may also increase seizure susceptibility and contribute to other aspects of brain injury pathophysiology (Danzer, 2018). Thus, it remains unclear what the overall effect of injury-induced neurogenesis following TBI might be, though its relevance to the overall adaptation for the hippocampus to traumatic brain injury is clear. We have added additional discussion around these points in the introduction (p3. lines 47 to 58).

2) Types of TBI are not addressed (in relation to APOE/TBI outcomes and TBI/neurogenesis effects).

We agree that we could make a clearer picture about the role of ApoE in human TBI as well as what is known about ApoE specifically around injury-induced neurogenesis. The ApoE4 allele is associated with small vessel disease and cognitive impairment on a vascular basis i.e. hypoxic-ischemic white matter damage and resulting dementia (Koizumi et al., 2018). ApoE4 is also associated with a greater incidence of moderate or severe contusions (Smith et al., 2006) as well as concussions (Merritt et al., 2018). Moreover, ApoE4 genotype combined with TBI is thought to increase the risk of developing Tauopathy and Alzheimer’s Disease (Cao et al., 2017) as well as post-traumatic epilepsy (Diaz-Arrastia et al., 2003) and impaired spontaneous blood brain barrier repair (Main et al., 2018). Neurogenesis is affected in an ApoE isoform-dependent manner after both ischemic stroke (Tobin et al., 2014) and controlled cortical impact (Hong et al., 2016) in mouse models but not after concussion (Wang et al., 2016). Recently a large meta-analysis in humans concluded that outcome following TBI occurs in an ApoE isoform dependent manner thus validating its importance in overall TBI but where the underlying mechanism remains entirely unknown (McFadyen et al., 2019. We have added additional discussion around these points in the introduction (p4. lines 80 to 89).

3) Effect of APOE4 in the context of TBI involves different mechanisms, which have been well explored molecularly in several studies. How much neurogenesis plays in the APOE4 dependent neurodegenerative or behavioral effects remains unknown?

We agree with the reviewer that it remains unclear how much neurogenesis plays in ApoE4-dependent behavioral deficits seen in any brain associated pathology. Our intent here is to highlight how injury-induced neurogenesis appears impaired in ApoE4 states in a manner that is similar to ApoE deficiency. Under normal conditions, mice lacking ApoE or expressing the E4 human allele have reduced neurogenesis and a smaller NSPC pool (Hong et al., 2016) as well as impaired dendritic maturation of surviving adult-born granule cell (Tensaouti et al., 2018). Moreover, injury-induced neurogenesis is greatly diminished in ApoE-deficient mice and decreased in ApoE4-expressing mice (Hong et al., 2016). Hippocampal-related cognitive performance is also impaired in healthy ApoE4 mice (Hartman et al., 2001; Peng et al., 2017; East et al., 2018) as well as in injured ApoE4 mice (Mannix et al., 2011; Teng et al., 2017). Furthermore, ApoE4 is the strongest genetic risk factor for late onset Alzheimer’s disease (Liu et al., 2013) where loss of neurogenesis has been linked with disease progression (Moreno-Jimenez et al., 2019). We have added in a number of references (noted above) both in the Introduction and Discussion to highlight these points.

4) Significant papers downplaying the role of neurogenesis in TBI are not referenced.

- Radoslaw Rola, Shinichiro Mizumatsu, Shinji Otsuka, Duncan R. Morhardt, Linda J. Noble-Haeusslein, Kelly Fishman, Matthew B. Potts, John R. Fike, Alterations in hippocampal neurogenesis following traumatic brain injury in mice, Volume 202, Issue 1, 2006, Pages 189-199, ISSN 0014-4886, https://doi.org/10.1016/j.expneurol.2006.05.034.

- R. Mark Richardson, Dong Sun, M. Ross Bullock, Neurogenesis After Traumatic Brain Injury, Neurosurgery Clinics of North America, Volume 18, Issue 1, 2007, Pages 169-181, ISSN 1042-3680, https://doi.org/10.1016/j.nec.2006.10.007.

- Kernie SG, Parent JM. Forebrain neurogenesis after focal Ischemic and traumatic brain injury. Neurobiol Dis. 2010;37(2):267–274. doi:10.1016/j.nbd.2009.11.002

We thank the reviewer for bringing these to our attention. We agree that it remains unclear exactly how important injury-induced neurogenesis might be following traumatic brain injury and based on the suggestion here, have now expanded on this point further and included the relevant citations (p3. lines 49 to 58). 

Reviewer #1: Design/Methods/results

5)No sham mice have been included, as the contralateral hemisphere is used as a control. This is problematic and needs to be addressed. As a result the authors focus on genotype effects in injured mice alone. The authors need to include sham mice (sham injuries/cranial window and injections only), or at least demonstrate in this manuscript that sham mice (ipsilateral hemisphere with cranial window) show comparable changes with the contralateral hemisphere of injured mouse brains. Additionally, data needs to be presented to show that there were no difference in the iGCL vs oGCL of the contralateral region.

We recently published the dendritic arborizations and spine density in uninjured dentate gyrus neurons in wildtype and various ApoE conditions (Tensaouti et al., 2018). We compared these to the contralateral neurons in injured animals analyzed in the present manuscript and found no significant differences, and therefore believe this serves as an adequate control without bringing in data from our already published work. We have now made this point clear in the materials and methods as a justification for using the contralateral side as a control (p10. lines 199 t0 204). One hallmark of TBI in the hippocampus is the increase of neurogenesis and the outward migration of adult born granule cells in the ipsilateral dentate gyrus only. We only observe very rare oGCL neuron on the contralateral side and are therefore unable to compare them to iGCL neurons and therefore are not able to do this analysis. This is predicted from what we and others have shown following unilateral traumatic injury (Villasana et al., 2015).

6) There is no mention of injury effects within each genotype for APOE KO, APOE3 and APOE4 mice (Figure 2 and 3, 4A-B). For example no changes in spine density (between ipsi vs contra) was observed in WT, KO, E3 and E4 mice. In this case, the changes in the contra hemisphere were similar to the changes in the ipsi hemisphere, showing that the effects are driven by genotype and not injury. This should be highlighted and discussed.

We have added 2 supplementary figures to describe the intra-condition comparisons of dendritic morphology to highlight the changes driven by the injury in each genotype. Both the dendritic complexity and the spine density of injury-induced adult-born granule neurons were similar in the 3 studied areas (iGCL/oGCL and Contralateral hemispheres) for all 4 genotypes, revealing no injury-driven effect for these parameters. Thus, the differences observed in dendritic morphology and spine density when comparing WT with KO or E3 with E4 were strictly driven by genotype and are now highlighted in the discussion (p21-22 lines 383-389 and p24. lines 449 to 458).

7) Data in Figure 1 are included in graphs/analyses of data in Figure 2 and 4 which is redundant and repetitive. It will be better to include the analyses of all groups on the same graph as in Fig 4E-G (This would be very busy but easier for the reader to compare across the different groups if clear labeling is used).

We thank the reviewer for the comment and agree that concentrating the data without being redundant makes the story clearer. In this revised version we completely reworked how we present the data and have added in new data that we believe improve the manuscript. We have enhanced Figure 1 with new data showing ApoE expression in the various mice in various ApoE expressing states (see response to #8 below). We have now combined the original Figure 2 and 4 into one consolidated Figure 2 and similarly with original Figures 3 and 4 have been condensed into one consolidated Figure 3. In addition, we made a variety of what we believe are useful intra-group comparisons in Supplementary Figures 1 and 2. While useful for making obvious comparisons, the data are redundant with that presented now in Figures 2 and 3 and we therefore believe based on this reviewer’s comments that the main points of the story are now highlighted in a less confusing albeit highly concentrated manner.

8) Rationale for including APOE-KO mice is not clear, as this is not relevant to humans (the main question is the influence of APOE3 vs APOE4). Moreover comparison of APOE-KO mice with WT also does not add much to our understanding as mice express a different form of APOE.

We agree that examining ApoE-deficient mice is not as relevant in humans but it does provide insight into basic mechanisms of what ApoE does in the setting of injury-induced neurogenesis. The purpose of this study was two-fold: 1) to examine ApoE in injury-induced neurogenesis and to 2) to examine how human isoforms of ApoE function in the setting of injury-induced neurogenesis. Interestingly, we observe that ApoE deficiency nearly phenocopies the substitution of mouse ApoE with human ApoE4, thereby strengthening the hypothesis that ApoE4 works more as negative regulator of hippocampal neurogenesis. This is consistent with our previous published observations regarding ApoE and neurogenesis but the underlying mechanisms remain unclear (Yang et al., 2011, Hong et al., 2016, Tensaouti et al., 2018). We have added additional information around this in the discussion (p24. lines 449 to 458). 

9) Why did the authors not collect cell counts in their analyses using the optical fractionator?

Retroviruses infect dividing cells stochastically, and therefore do not infect all dividing cells that are present in the area injected. Therefore, using classical quantification techniques such as with an optical fractionator are not accurate. The purpose of this particular study was to quantitatively assesses the characteristics of the dendritic arborizations and spine density, which are not dependent on quantitative cell counts. Since we have previously published quantitative cell counts of injury-induced neurogenesis in various ApoE states and do not feel it would add anything to the present story.

10) It was stated that astrocytes drive NSPCs to astrogenesis, it would add value to the paper if the authors obtained double staining for differentiated astrocytes as they did in their previous paper with nestin, ki-67 and GFAP etc... and APOE (Tensaouti et al., 2018).

We appreciate the suggestion and we have added in new data as suggested with a revised Figure 1 where we now show immunostaining for ApoE, GFAP and GFP in the dentate gyrus from the 4 genotypes analyzed.

11) If mice were administered Brdu, it would be good to include Brdu/Prox1 cell density in the different granular layers.

We did not administer Brdu to these animals as the work alluded to has been done by us previously and the focus here was in quantifying neuron development with Sholl analysis and spine density. 

12) 200: Says "mixed genotype and....." however the legend states that the data were from WT mice.

Thank you for bringing this to our attention, the text has been corrected accordingly.

13) What was the rationale for injuries at 6 weeks, this is still very young and the brain is still developing in mice.

This is the timepoint we have chosen for most of our studies involving dentate gyrus neurogenesis as, although it is young in mice, it is a timepoint known to be well past the early childhood equivalent in humans. Moreover, the neurons were not analyzed until 10 weeks of age which is well into “early adulthood” of mice.

14) Discuss in some more detail, the rationale for the injection at 4 weeks post-injury. How does this relate to the timeline of events after CCI, and what could be missed from this timepoint of analyses.

Injections were done immediately after injury, but brains were analyzed 4 weeks later. We choose this 4-week timepoint deliberately, because it is the time necessary for an adult born granule cell to become fully mature and integrated to the pre-existing circuitry (Toni and Schinder, 2015). Here, we are interested in studying injury-induced adult-born granule cells that survived and matured and have integrated into the existing circuitry. These aspects of development are separated from peaks of proliferation, neural degeneration and inflammation, which occur much sooner (Zhou et al., 2012; Taib et al., 2017) as does the speed of dendritic development (Villasana et al., 2015). 

15) 211: is there another term that could be used instead of "spread" of adult granule neurons.

As suggested, we have changed this to dendritic span.

16) Reviewer #1: Discussion: Their discussion needs to be expanded and re-written based on the comments above.

We appreciate this reviewer’s thoughtful and through critique. In addition to the specific points outlined in 1-15 above, the discussion has been reworked and expanded as suggested. Additions are now visible in blue.

Reviewer #2

In the manuscript, the authors investigated the effects of ApoE gene in the regulation of the maturation of injury-induced adult-born neurons in the hippocampus following traumatic brain injury. Overall, the manuscript is interesting and contributes to the understanding of ApoE genes in the regulation of adult hippocampal neurogenesis after brain injury. However, some concerns need to be addressed.

1) In the study, mixed sex mice were used as the authors claimed. What is the percentage of each sex mice in each group? Are there the same numbers of female and male mice investigated in each genotype? It is known that there is sex difference in basal adult neurogenesis, however, the authors did not mention the numbers of male and female mice in each group they studied.

Thank you for bringing this to our attention, we used a total of 16 animals, 4 in each group, with 2 males and 2 females in each case. The manuscript has been updated accordingly (p6, lines 110 to 111).

2) The authors should provide more information about the retrovirus carrying eGFP they used in the study, it will help the readers understand the work more easily

.

Retroviruses, as opposed to adenoviruses which infect post-mitotic cells and lentiviruses which inject both post-mitotic and mitotic cells, can only infect mitotic (actively dividing) cells. As a result, only neural stem and progenitor cells from the neuronal lineage can express GFP in the infected dentate gyrus. Other potentially dividing cells such as reactive astrocytes and microglia can morphologically be differentiated without ambiguity. Therefore, we chose to use a GFP-expressing retrovirus so that we can capture the progenitor cells when they are dividing and follow their development, which takes 3-4 weeks before they are fully mature, hence our choosing this timeframe after injection. We have made these points clearer in the materials and methods and results (see p7, lines 128 to 131).

3) In result2, the authors claimed that they found significant interaction in both the iGCL and oGCL. However, in table1, the results shows the interaction of sholl analysis in oGCL is not significant.

Thank you for bringing this to our attention, this has been corrected.

4) In the two-way ANOVA they used to analyze the sholl anaylasis results, they provided the details of the results. However, what is the row factor in the analysis? They should specifically point out what is row factor and what is column factor.

The Sholl Analysis represents the average number of intersections in each condition (Column) function of the distance to the soma (Row). We have updated the statistic table legend (p11, line 213).

5) In result4, they claimed that "the reduced spine density in ApoE-KO mice adult-born neurons is exaggerated after CCI". However, the result of the ApoE-KO adult born neurons without CCI is published in another paper. I am wondering if the authors conduct the two studies by using the mice purchased at the same time and performed the studies at the same time? If not, it is not very appropriate to compare these two results.

The two studies began at the same time and were conducted largely simultaneously and in the exact same conditions; only ApoE4 mice were purchased specially for the experiments, while WT, ApoE KO and ApoE3 mice are continuously generated by our lab. We have completely reworked the figures as suggested by Reviewer 1 to hopefully clarify these points and not to present redundant data. Further comparisons that do present some of these redundant data are now included in Supplementary Figures 1 and 2.

6) More details about the ApoE-KO, ApoE3 and ApoE4 mice will help the readers understand the manuscript better.

Thank you for the suggestion and a number of details about these mice have been added to the discussion including (in summary):

It has been shown that mice lacking ApoE or expressing ApoE4 exhibit cognitive impairments (Fuentes et al., 2018; Oitzl et al., 1997) (Masliah et al. 1997); (Grootendorst et al. 2001); (Peister et al. 2006); (Rodriguez et al. 2013); (Salomon-Zimri et al. 2015); (Peng et al. 2017); (East et al. 2018) while ApoE3 and WT mice are similar, particularly in spatial learning/memory and in olfactory memory, two neurogenic-dependent behaviors (p25. lines 449 to 458).

REFERENCES

1. Gao X, Deng-Bryant Y, Cho W, Carrico KM, Hall ED, Chen J. Selective death of newborn neurons in hippocampal dentate gyrus following moderate experimental traumatic brain injury. J Neurosci Res. 2008;86(10):2258-70.

2. Yu TS, Zhang G, Liebl DJ, Kernie SG. Traumatic brain injury-induced hippocampal neurogenesis requires activation of early nestin-expressing progenitors. J Neurosci. 2008;28(48):12901-12.

3. Villasana LE, Kim KN, Westbrook GL, Schnell E. Functional Integration of Adult-Born Hippocampal Neurons after Traumatic Brain Injury(1,2,3). eNeuro. 2015;2(5):ENEURO.0056-15.2015. Published 2015 Sep 28. doi:10.1523/ENEURO.0056-15.2015

4. Blaiss CA, Yu TS, Zhang G, Chen J, Dimchev G, Parada LF, et al. Temporally specified genetic ablation of neurogenesis impairs cognitive recovery after traumatic brain injury. J Neurosci. 2011;31(13):4906-16

5. Danzer SC. Contributions of Adult-Generated Granule Cells to Hippocampal Pathology in Temporal Lobe Epilepsy: A Neuronal Bestiary. Brain Plast. 2018;3(2):169-81.

6. Koizumi, K., Hattori, Y., Ahn, S.J. et al. Apoε4 disrupts neurovascular regulation and undermines white matter integrity and cognitive function. Nat Commun 9, 3816 (2018) doi:10.1038/s41467-018-06301-2

7. Smith C, Graham DI, Murray LS, Stewart J, Nicoll JA. Association of APOE e4 and cerebrovascular pathology in traumatic brain injury. J Neurol Neurosurg Psychiatry. 2006;77(3):363–366. doi:10.1136/jnnp.2005.074617

8. Victoria C Merritt, Amanda R Rabinowitz, Peter A Arnett, The Influence of the Apolipoprotein E (APOE) Gene on Subacute Post-Concussion Neurocognitive Performance in College Athletes, Archives of Clinical Neuropsychology, Volume 33, Issue 1, February 2018, Pages 36–46, https://doi.org/10.1093/arclin/acx051

9. Cao, J., Gaamouch, F.E., Meabon, J.S. et al. ApoE4-associated phospholipid dysregulation contributes to development of Tau hyper-phosphorylation after traumatic brain injury. Sci Rep 7, 11372 (2017) doi:10.1038/s41598-017-11654-7

10. Diaz-Arrastia R, Gong Y, Fair S, et al. Increased Risk of Late Posttraumatic Seizures Associated With Inheritance of APOE ϵ4 Allele. Arch Neurol. 2003;60(6):818–822. doi:https://doi.org/10.1001/archneur.60.6.818

11. Main, B.S., Villapol, S., Sloley, S.S. et al. Apolipoprotein E4 impairs spontaneous blood brain barrier repair following traumatic brain injury. Mol Neurodegeneration 13, 17 (2018) doi:10.1186/s13024-018-0249-5

12. Tobin MK, Bonds JA, Minshall RD, Pelligrino DA, Testai FD, Lazarov O. Neurogenesis and inflammation after ischemic stroke: what is known and where we go from here. J Cereb Blood Flow Metab. 2014;34(10):1573–1584. doi:10.1038/jcbfm.2014.130

13. Hong S, Washington PM, Kim A, Yang CP, Yu TS, Kernie SG. Apolipoprotein E Regulates Injury-Induced Activation of Hippocampal Neural Stem and Progenitor Cells. J Neurotrauma. 2016;33(4):362–374. doi:10.1089/neu.2014.3860

14. Wang X, Gao X, Michalski S, Zhao S, Chen J. Traumatic Brain Injury Severity Affects Neurogenesis in Adult Mouse Hippocampus. J Neurotrauma. 2016;33(8):721–733. doi:10.1089/neu.2015.4097

15. Tensaouti Y, Stephanz EP, Yu TS, Kernie SG. ApoE Regulates the Development of Adult Newborn Hippocampal Neurons. eNeuro. 2018;5(4).

16. R.E. Hartman, D.F. Wozniak, A. Nardi, et al.. Behavioral phenotyping of GFAP-apoE3 and -apoE4 transgenic mice: apoE4 mice show profound working memory impairments in the absence of Alzheimer's-like neuropathology. Exp Neurol, 170 (2) (2001), pp. 326-344

17. Peng KY, Mathews PM, Levy E, Wilson DA. Apolipoprotein E4 causes early olfactory network abnormalities and short-term olfactory memory impairments. Neuroscience. 2017;343:364–371. doi:10.1016/j.neuroscience.2016.12.004 

18. East BS, Fleming G, Peng K, et al. Human Apolipoprotein E Genotype Differentially Affects Olfactory Behavior and Sensory Physiology in Mice. Neuroscience. 2018;380:103–110. doi:10.1016/j.neuroscience.2018.04.009 

19. Mannix RC, Zhang J, Park J, et al. Age-dependent effect of apolipoprotein E4 on functional outcome after controlled cortical impact in mice. J Cereb Blood Flow Metab. 2011;31(1):351–361. doi:10.1038/jcbfm.2010.99 

20. Teng, Z., Guo, Z., Zhong, J. et al. ApoE Influences the Blood-Brain Barrier Through the NF-κB/MMP-9 Pathway After Traumatic Brain Injury . Sci Rep 7, 6649 (2017) doi:10.1038/s41598-017-06932-3

21. Liu CC, Liu CC, Kanekiyo T, Xu H, Bu G. Apolipoprotein E and Alzheimer disease: risk, mechanisms and therapy [published correction appears in Nat Rev Neurol. 2013. doi: 10.1038/nmeurol.2013.32. Liu, Chia-Chan [corrected to Liu, Chia-Chen]]. Nat Rev Neurol. 2013;9(2):106–118. doi:10.1038/nrneurol.2012.263

22. Moreno-Jiménez, E.P., Flor-García, M., Terreros-Roncal, J. et al. Adult hippocampal neurogenesis is abundant in neurologically healthy subjects and drops sharply in patients with Alzheimer’s disease. Nat Med 25, 554–560 (2019) doi:10.1038/s41591-019-0375-9

23. Radoslaw Rola, Shinichiro Mizumatsu, Shinji Otsuka, Duncan R. Morhardt, Linda J. Noble-Haeusslein, Kelly Fishman, Matthew B. Potts, John R. Fike, Alterations in hippocampal neurogenesis following traumatic brain injury in mice, Volume 202, Issue 1, 2006, Pages 189-199, ISSN 0014-4886, https://doi.org/10.1016/j.expneurol.2006.05.034.

24. R. Mark Richardson, Dong Sun, M. Ross Bullock, Neurogenesis After Traumatic Brain Injury, Neurosurgery Clinics of North America, Volume 18, Issue 1, 2007, Pages 169-181, ISSN 1042-3680, https://doi.org/10.1016/j.nec.2006.10.007.

25. Kernie SG, Parent JM. Forebrain neurogenesis after focal Ischemic and traumatic brain injury. Neurobiol Dis. 2010;37(2):267–274. doi:10.1016/j.nbd.2009.11.002

26. Yang CP, Gilley JA, Zhang G, Kernie SG. ApoE is required for maintenance of the dentate gyrus neural progenitor pool. Development. 2011;138(20):4351-62.

27. Toni N, Schinder AF. Maturation and Functional Integration of New Granule Cells into the Adult Hippocampus. Cold Spring Harb Perspect Biol. 2015;8(1):a018903.

28. Zhou H, Chen L, Gao X, Luo B, Chen J. Moderate traumatic brain injury triggers rapid necrotic death of immature neurons in the hippocampus. J Neuropathol Exp Neurol. 2012;71(4):348–359. doi:10.1097/NEN.0b013e31824ea078

29. Taib T, Leconte C, Van Steenwinckel J, et al. Neuroinflammation, myelin and behavior: Temporal patterns following mild traumatic brain injury in mice. PLoS One. 2017;12(9):e0184811. Published 2017 Sep 14. doi:10.1371/journal.pone.0184811

30. Fuentes D, Fernández N, García Y, García T, Morales AR, Menéndez R. Age-Related Changes in the Behavior of Apolipoprotein E Knockout Mice. Behav Sci (Basel). 2018;8(3):33. Published 2018 Mar 3. doi:10.3390/bs8030033

31. Melly S. Oitzl, Monique Mulder, Paul J. Lucassen, Louis M. Havekes, Jeannette Grootendorst, E.Ron de Kloet, Severe learning deficits in apolipoprotein E-knockout mice in a water maze task, Brain Research, Volume 752, Issues 1–2, 1997, Pages 189-196, ISSN 0006-8993, https://doi.org/10.1016/S0006-8993(96)01448-5.Masliah E, Samuel W, Veinbergs I, Mallory M, Mante M, Saitoh T. 1997. Neurodegeneration and cognitive impairment in apoe-deficient mice is ameliorated by infusion of recombinant apoe. Brain Res. 751(2):307-314.

32. Grootendorst J, de Kloet ER, Dalm S, Oitzl MS. 2001. Reversal of cognitive deficit of apolipoprotein e knockout mice after repeated exposure to a common environmental experience. Neuroscience. 108(2):237-247.

33. Peister A, Zeitouni S, Pfankuch T, Reger RL, Prockop DJ, Raber J. 2006. Novel object recognition in apoe(-/-) mice improved by neonatal implantation of wild-type multipotential stromal cells. Exp Neurol. 201(1):266-269.

34. Rodriguez GA, Burns MP, Weeber EJ, Rebeck GW. 2013. Young apoe4 targeted replacement mice exhibit poor spatial learning and memory, with reduced dendritic spine density in the medial entorhinal cortex. Learn Mem. 20(5):256-266.

35. Salomon-Zimri S, Liraz O, Michaelson DM. 2015. Behavioral testing affects the phenotypic expression of apoe ε3 and apoe ε4 in targeted replacement mice and reduces the differences between them. Alzheimers Dement (Amst). 1(2):127-135.

36. Peng KY, Mathews PM, Levy E, Wilson DA. 2017. Apolipoprotein e4 causes early olfactory network abnormalities and short-term olfactory memory impairments. Neuroscience. 343:364-371.

37. East BS, Fleming G, Peng K, Olofsson JK, Levy E, Mathews PM, Wilson DA. 2018. Human apolipoprotein e genotype differentially affects olfactory behavior and sensory physiology in mice. Neuroscience. 380:103-110.

38. E. Masliah, W. Samuel, I. Veinbergs, M. Mallory, M. Mante, T. Saitoh. Neurodegeneration and cognitive impairment in apoE-deficient mice is ameliorated by infusion of recombinant apoE. Brain Res., 751 (1997), pp. 307-314

---

## [Decision Letter · Decision Letter 1]

3 Feb 2020

Apolipoprotein E regulates the maturation of injury-induced adult-born hippocampal neurons following traumatic brain injury

PONE-D-19-27231R1

Dear Dr. Kernie,

We are pleased to inform you that your manuscript has been judged scientifically suitable for publication and will be formally accepted for publication once it complies with all outstanding technical requirements.

With kind regards,

Stephen D. Ginsberg, Ph.D.

Section Editor

PLOS ONE

**Comments to the Author**

1. If the authors have adequately addressed your comments raised in a previous round of review and you feel that this manuscript is now acceptable for publication, you may indicate that here to bypass the “Comments to the Author” section, enter your conflict of interest statement in the “Confidential to Editor” section, and submit your "Accept" recommendation.

Reviewer #1: All comments have been addressed

Reviewer #2: All comments have been addressed

2. Is the manuscript technically sound, and do the data support the conclusions?

Reviewer #1: Yes

Reviewer #2: Yes

3. Has the statistical analysis been performed appropriately and rigorously? 

Reviewer #1: Yes

Reviewer #2: Yes

4. Have the authors made all data underlying the findings in their manuscript fully available?

Reviewer #1: Yes

Reviewer #2: Yes

5. Is the manuscript presented in an intelligible fashion and written in standard English?

Reviewer #1: Yes

Reviewer #2: Yes

6. Review Comments to the Author

Reviewer #1: The authors have addressed the comments adequately and made substantial corrections to the original submission.

Reviewer #2: (No Response)

7. PLOS authors have the option to publish the peer review history of their article (what does this mean?). If published, this will include your full peer review and any attached files.

Reviewer #1: Yes: Joseph O Ojo

Reviewer #2: Yes: Hao Wang

---

## [Editor Report · Acceptance letter]

10 Feb 2020

PONE-D-19-27231R1 

Apolipoprotein E regulates the maturation of injury-induced adult-born hippocampal neurons following traumatic brain injury 

Dear Dr. Kernie:

I am pleased to inform you that your manuscript has been deemed suitable for publication in PLOS ONE. Congratulations! Your manuscript is now with our production department. 

With kind regards,

on behalf of

Dr. Stephen D. Ginsberg 

Section Editor

PLOS ONE